# Post-CDK 4/6 Inhibitor Therapy: Current Agents and Novel Targets

**DOI:** 10.3390/cancers15061855

**Published:** 2023-03-20

**Authors:** Nadia Ashai, Sandra M. Swain

**Affiliations:** Department of Medicine, Georgetown Lombardi Comprehensive Cancer Center and MedStar Health, Washington, DC 20007, USA

**Keywords:** metastatic breast cancer, CDK4/6 inhibitor, hormone receptor positive, SERD, HER2 low, second line

## Abstract

**Simple Summary:**

CDK4/6 inhibitors (CDK4/6i) with endocrine therapy are the established first-line treatment for metastatic and advanced hormone receptor-positive breast cancer (mBC). Recently, there has been an expansion in available next lines of therapy; however, optimal sequencing remains unclear. This paper reviews data on the efficacy and response rates of approved therapeutic options, and when possible, discusses their efficacy in the setting of prior exposure to CDK4/6i. This paper also seeks to review emerging targets and therapeutics that may be approved in the future for this patient population.

**Abstract:**

Front-line therapy for advanced and metastatic hormone receptor positive (HR+), HER2 negative (HER−) advanced or metastatic breast cancer (mBC) is endocrine therapy with a CDK4/6 inhibitor (CDK4/6i). The introduction of CDK4/6i has dramatically improved progression-free survival and, in some cases, overall survival. The optimal sequencing of post-front-line therapy must be personalized to patients’ overall health and tumor biology. This paper reviews approved next lines of therapy for mBC and available data on efficacy post-progression on CDK4/6i. Given the success of endocrine front-line therapy, there has been an expansion in therapies under clinical investigation targeting the estrogen receptor in novel ways. There are also clinical trials ongoing attempting to overcome CDK4/6i resistance. This paper will review these drugs under investigation, review efficacy data when possible, and provide descriptions of the adverse events reported.

## 1. Introduction

Breast cancer is the most common cancer diagnosed globally. In the United States, between 2014 and 2018, the incidence of breast cancer increased by 0.5% annually [1]. Early detection due to screening efforts and improved treatment options for localized disease has decreased breast cancer mortality. It is estimated that 4 million women in the United States are living with breast cancer, and more than 150,000 of those women are living with metastatic disease [2,3]. While 5-year survival rates for localized breast cancer are 99%, it is limited to 29% for patients diagnosed with metastatic disease [4]. Hormone receptor-positive (HR+) breast cancer remains the most common type of breast cancer diagnosed in the United States. It is estimated that 70% of all new breast cancer cases are HR+. These cancers are marked by the ability to be treated with hormonally targeted therapy.

CDK4/6 inhibitors (CDK4/6i) with endocrine therapy are the standard of care for HR+, HER2−negative (HER−) mBC in the front-line setting. CDK4/6i inhibits cell cycle progression by inhibiting the downstream effects of the CDK4/6 complex with cyclin D. Cyclin D amplification is common in HR+ breast cancer. Cyclin D and CDK4/6 proteins form a complex that promotes mitosis through various pathways. This complex inhibits tumor suppressor RB1, which in turn promotes the proliferation of cyclins involved in the DNA replication and transcription of FOXM1, which in turn supports the G2/M phases of the cell cycle. The addition of CDK 4/6i to an endocrine backbone has improved progression-free survival (PFS) and overall survival (OS) in pre- and postmenopausal women.

There are three Food and Drug Administration (FDA)-approved CDK4/6i. Palbociclib (Ibrance), ribociclib (Kisqali), and abemaciclib (Verzenio) had remarkably similar PFS in PALOMA-2, MONALEESA-2, and MONARCH-3, which were trials of 24.8 months, 25.3 months, and 28.2 months, respectively [5,6,7]. In the first-line setting, ribociclib combined with endocrine therapy showed an overall survival (OS) benefit [8,9]. Abemaciclib, at its second interim analysis, showed numerical improvement in OS; however, it has not reached statistical significance [10]. PALOMA-2, which followed patients on palbociclib with endocrine therapy, was unable to demonstrate an overall survival benefit in comparison to endocrine therapy alone [11]. It remains unclear if this is due to patients being lost to follow up, limiting adverse events (AE), or because of differences in drug efficacy.

While the addition of CDK4/6i to endocrine therapy in the front-line setting has been a needed advancement for patients with advanced HR+ breast cancer, there is no standard of care for second-line therapy. Most data on second-line therapies were obtained prior to the widespread use of a CDK4/6i. This article will review second- and third-line therapy options and their efficacy post-CDK4/6i. We will also review options for sequencing and review novel agents under investigation for patients with HR+ mBC.

## 2. Second-Line Therapy Options

### 2.1. Poly (ADP-Ribose) Polymerase Inhibitors

Oral poly (ADP-ribose) polymerase inhibitors (PARPi) are approved by the FDA for use in patients with germline BRCA mutations (gBRCA) in the metastatic setting. While data suggest promising response rates for patients with germline PALB2 mutations and somatic BRCA mutations, studies are needed to confirm their benefit [12,13]. In patients with gBRCA and advanced breast cancer, olaparib (Lynparza) improved PFS in comparison to chemotherapy (7.0 months vs. 4.2 months; HR 0.58; 95% CI, 0.43–0.80; *p* < 0.001) [14]. There was a suggestion of an overall survival (OS) benefit in patients who received PARPi as front-line therapy in a patient population that included patients with HR+, HER2− disease (HR 0.51, 95% CI 0.29–0.90); however, this benefit was not seen in patients who used olaparib in the second or third line [15]. Given the remarkable PFS with front-line CDK4/6i, PARPi are not recommended as first-line therapy for HR+, HER2− mBC. They are currently FDA-approved for use in first- to third-line therapy and, whenever possible, should be considered prior to the use of chemotherapy.

Currently, no data are available on PARPi efficacy post-CDK4/6 therapy; however, studies suggest that patients with germline BRCA mutations have poorer responses to initial CDK4/6i therapy. A retrospective analysis of patients with and without germline DNA repair mutations (gBRCA1/2-ATM-CHEK2) were evaluated for their response to CDK4/6i treatment. Of the 70 patients that were treated with CKD4/6i as first-line therapy, median PFS was shorter in patients with germline mutations compared to wild-type and non-tested patients (10.2 months, 15.6 months, and 17.6 months, respectively; *p* = 0.002) [16].

In another retrospective analysis, patients with BRCA2 germline mutations had a poorer PFS on CDK4/6i than patients with wild-type tumors. Wild-type BRCA2 had a PFS of 14.7 months vs. germline-mutated BRCA2 with a PFS of 7.0 months (HR 2.32, 1.38–3.91, *p* = 0.0016). Of the 122 patients with germline BRCA2 mutations, tumor mutational analysis showed the somatic enrichment of MYC and RB1 [17]. This is significant, as RB1 overexpression is a suggested mechanism of CDK4/6i resistance. Consideration is being given to prospectively evaluating the safety and efficacy of combining CDK4/6i and PARPi due to the synergistic effects seen in vitro and in vivo [18].

### 2.2. Alpelisib

Alpelisib is a PIK3CA kinase inhibitor. Targeted therapy with alpelisib in combination with fulvestrant is an important second- or third-line option for patients with HR+, HER2− mBC with PIK3CA-mutated tumors. PICK3CA somatic mutations are present in approximately 40% of patients with mBC. Alpelisib prolonged PFS in comparison to placebo and fulvestrant in patients who have progressed on endocrine therapy in the Phase III SOLAR-1 trial. Progression-free survival on alpelisib compared to fulvestrant monotherapy was 11.0 months vs. 5.7 months, respectively (HR for progression or death, 0.65; 95% CI, 0.50–0.85; *p* < 0.001). In this trial, however, only 5.9% of patients (*n* = 20) had prior exposure to CDK4/6 inhibition [19]. Unfortunately, alpelisib and fulvestrant was associated with an OS of 39.3 months compared to fulvestrant monotherapy with its OS of 31.4 months, which was not statistically significant (HR 0.86, 95% CI, 0.64–1.15, *p* = 0.15) [20].

In subsequent studies, alpelisib and fulvestrant have shown impressive PFS benefit immediately post-CDK4/6i therapy (Table 1). In cohort A of the Phase II open-label BYLieve trial, patients received alpelisib and fulvestrant immediately post front-line therapy with CK4/6i and an aromatase inhibitor (AI). The median PFS in this cohort was 7.3 months, making this intervention an important second- or third-line option for patients whose tumors have a PIK3CA mutation and have progressed on a CDK4/6i [21]. Of note, the mPFS in cohort A with ESR1 mutations was 5.6 months compared to 8.3 months in the WT population. While patients with ESR1-mutated tumors had a numerically worse PFS, this difference was not statistically significant. Interestingly, in Cohort B, patients on alpelisib with an AI had a statistically significantly worse mPFS in the presence of an ESR1 mutation, further suggesting that ESR1 mutations cause resistance to Ais.

In BYLieve Cohort C, patients received alpelisib and fulvestrant and were eligible if they had progressed on AI and received chemotherapy or ET as immediate prior treatment. Eighty four percent of patients in this cohort had prior treatment with a CDK4/6i in the metastatic setting. The median PFS was 5.6 months, with an overall response rate (ORR) of 28% and a clinical benefit rate (CBR) of 43% in the intention-to-treat population. Interestingly, the median PFS was higher at 6.4 months in the population that immediately had CDK4/6i as their prior treatment [22]. In patients who had been treated with a prior CDK4/6i and harbored a PIK3CA mutation, alpelisib with fulvestrant should be considered as a second- or third-line therapy given its impressive PFS. The benefits of this therapy should be weighed against toxicities, primarily hyperglycemia.

### 2.3. Fulvestrant

Fulvestrant is a selective estrogen receptor degrader delivered intramuscularly, and it is an important monotherapy option for patients with HR+, HER− mBC. The real-world efficacy of fulvestrant monotherapy as a first-line endocrine therapy has shown a durable PFS of between 12 and 16 months [35,36]. In the SoFEA trial, fulvestrant with or without an additional endocrine therapy had a median PFS of 3.4–4.8 months after a nonsteroidal aromatase inhibitor (AI). Median PFS did not differ between monotherapy fulvestrant or in combination with an AI [37]. The benefit of fulvestrant monotherapy was confirmed in another Phase III, multicenter blinded study, which showed a median duration of clinical benefit of 9.3 months post-progression on a nonsteroidal AI [38]. There is a noticeable decline in PFS benefit when comparing front-line and second-line monotherapy fulvestrant trials, which suggests progressive endocrine resistance limiting the use of fulvestrant’s efficacy in later lines.

Fulvestrant monotherapy has been evaluated as a control arm in two prospective trials post-CDK4/6i use. In the Phase III EMERALD study, all patients had progressed on 1–2 lines of endocrine therapy, including one in combination with a CDK4/6i. The fulvestrant monotherapy control arm had a median PFS of 1.9 months (95% CI 1.87–2.1). Patients had a 6-month PFS rate of 22.9% and a 12-month PFS rate of 10.2% [39]. In VERONICA, fulvestrant monotherapy was also associated with a clinical benefit rate of 13.7% and a PFS of 1.94 months (95% CI 1.84–3.55), suggesting a limited utility of this option post-front-line endocrine therapy [24].

While fulvestrant monotherapy may have limited utility post-CDK4/6i, it remains an important backbone of therapy in combination with other drugs. When given with alpelisib, it can overcome endocrine resistance in patients with ESR1 mutations, as discussed above. It also can be given with mTOR inhibitors, such as everolimus. Of note, fulvestrant and everolimus post-progression on endocrine therapy had an mPFS of 12.3 months PFS (95% CI, 7.7–15.7 months) in the Phase II MANTA trial. Fulvestrant monotherapy had a PFS of 5.4 months (95% CI, 3.5–9.2 months). Fulvestrant with everolimus had more Grade 3 or 4 events compared to fulvestrant monotherapy, most significantly stomatitis (11.7% vs. 0%) and infection (6.7% vs. 0%), respectively. This trial completed accrual prior to the widespread use of CDK4/6i; however, it does suggest that fulvestrant paired with an mTOR inhibitor is a safe option to consider for patients in the second-line setting [40]. Recently, elacestrant has received FDA approval in patients with prior endocrine therapy history and ESR1-mutated tumors. In patients without this biomarker, fulvestrant remains a well-tolerated endocrine therapy option. These authors suggest close radiographic surveillance on monotherapy fulvestrant post-CDK4/6i, as PFS benefit post-CDK4/6i was limited to under 2 months in two Phase III clinical trials.

Of note, fulvestrant may have improved efficacy when selected based on molecular profiling prior to the progression of the disease. In PADA-1, patients receiving an AI with palbociclib had ctDNA monitoring for an ESR1 mutation at inclusion, one month, and then every two months. If they were confirmed to have an ESR1 mutation without progression of disease, they were randomized to continuing therapy or changing to fulvestrant and palbociclib. After randomization, patients on palbociclib and an AI had a mPFS of 5.7 months; however, patients on palbociclib and fulvestrant had an mPFS of 11.9 months. This difference was statistically significant with a stratified HR of 0.61 (95% CI 0.43–0.86, *p* = 0.004) [41]. While these results are promising, it remains unclear if survival would be altered with changing therapy at ESR1 mutation development versus at radiographic progression. It also remains unclear if this strategy will remain relevant post-FDA approval of elacestrant.

### 2.4. Exemestane and Everolimus

Everolimus inhibits the mammalian target of rapamycin (mTOR) pathway, which is implicated in cell proliferation and a mechanism of endocrine resistance. Everolimus and exemestane is an established therapy option for HR+ HER2− mBC that may also have efficacy post-CDK4/6i. In a retrospective study of 41 patients who had progressed on palbociclib, patients had an mPFS of 4.2 months (95% CI 3.2–6.2) with a median OS of 18.7 months on everolimus and exemestane. ORR and CBR were both 17.1% [25]. Another analysis of real-world outcomes on exemestane and everolimus post-CDK4/6i from the Alberta Health Services evaluated 20 patients who were on everolimus and exemestane after being on a CDK4/6i in the first or second line. Median time on hormonal therapy with a CDK4/6i was 12.0 months (range 4.0–20.9 months). Median PFS on everolimus and exemestane was 5.8 months [26].

In another retrospective analysis at a single institution, patients treated with at least one cycle of exemestane and everolimus who had previously progressed on AI vs. AI and CDK4/6i were compared [27]. Seventeen patients who had progressed on a CDK4/6i had a median PFS of 3.6 months on exemestane and everolimus, compared to 4.2 months in those who had not been on a CK4/6i previously. This difference was not statistically significant. Patients were on CDK4/6i for a median of 10.3 months (2.8–33.4). At 6 months, 17.6% of patients with prior CDK4/6i exposure remained on everolimus and exemestane.

In another retrospective analysis from Osaka International Cancer Institute, 12 patients received palbociclib followed by exemestane and everolimus [28]. They had up to 0–6 lines of therapy prior to palbociclib. Patients were on CDK4/6i for a median of 5 months and remained on mTOR inhibitors for a median of 11.7 months. While these studies are small, it is notable that PFS durations widely vary post-CDK4/6i exposure, and mTOR inhibitors may be more effective in patients with shorter durations of response to CDK4/6i (Table 1). Additional studies are needed to better characterize PFS expectations post-CDK4/6i with information regarding ESR1 mutations, which may result in a shorter PFS.

### 2.5. Trastuzumab Deruxtecan

Trastuzumab deruxtecan (T-DXd) is an antibody drug conjugate consisting of monoclonal antibody trastuzumab with a topoisomerase I inhibitor payload. Results of the Phase III DESTINY-Breast04 trial established a new breast cancer treatment classification of HR+, HER2 low disease. HER2 low disease is defined as patients with a HER2 IHC of 1–2 plus without gene amplification. In DESTINY-Breast04, patients with HR-positive and -negative disease were randomized in a 2:1 fashion to T-DXd versus physician’s choice of chemotherapy. Patients on trial had received 1–2 lines of chemotherapy, and patients with HR+ disease were required to have endocrine resistant disease. Within the HR+ patient population, 78% of patients had a prior CDK4/6i in the investigational arm, and in the control arm, 81% of patients had had a prior CDK4/6i. The primary endpoint of the trial was PFS by blinded independent central review in patients with HR+ disease. Secondary endpoints included PFS benefit in all patients and OS. Of the 557 patients enrolled, 89% of patients had HR-positive disease.

In the HR+ group, patients had an mPFS of 10.1 months on T-DXd versus 5.4 months in the control arm (HR 0.5164 95% CI 0.40–0.64, *p* < 0.0001). Patients also had a statistically significant improvement in mOS of 6.4 months (HR 0.64 95% CI 0.48–0.86, *p* = 0.0028). In the T-DXd arm, OS was 23.9 months versus 17.5 months in the control arm. Patients with HR+ disease had an ORR of 52.6% and a clinical benefit rate (CBR) of 71.2%, establishing T-DXd as a new standard of care post-CDK4/6i [30]. While patients on trial all had prior chemotherapy exposure, the OS benefit of 6.4 months noted in this trial suggests that using it earlier in sequencing may be prudent once patients develop endocrine-resistant disease, especially in patients with rapidly progressing disease or those that may be intolerant of chemotherapy. T-DXd prior to chemotherapy is currently being investigated in DESTINY-06. Consideration should be given to potential toxicities, including cytopenias, gastrointestinal upset, and the 12% incidence of interstitial lung disease (ILD). Of the patients that developed ILD, 5 patients (1.3%) had Grade 3 events, and 3 patients (0.8%) had Grade 5 events.

### 2.6. Sacituzumab Govitecan

Sacituzumab govitecan (SG) is an antibody drug conjugate that targets human trophoblast cell surface antigen 2 (TROP-2) and is designed to effectively deliver a chemotherapeutic agent. It initially received FDA approval for patients with triple-negative mBC. TROP-2 is overexpressed in other breast subtypes and has been studied in patients with HR+, HER− mBC.

In a Phase I/II, single-arm trial of patients with HR+, HER− mBC whose disease progressed on at least one prior endocrine-based therapy and chemotherapy, these patients were then followed on SG. Of 54 patients, 32 (59.3%) had previously been on a CDK4/6i. In the intention-to-treat population (ITT), patients had an ORR of 17% and a CBR of 24% compared to an ORR of 8% and CBR of 12 % in patients who had prior CDK4/6i. Patients who had prior CKD4/6i had a median PFS and OS of 3.8 months (95% CI 1.9–6.5) and 11.0 months (95% CI 8.1–16.4), respectively. In the ITT population, median PFS was 5.5 months (95% CI 3.6–7.6), with a median OS of 12.0 months (95% CI 9.0–18.2). Median PFS and OS were numerically longest in patients who had not received CDK4/6i at 7.6 months (95% CI 5.1–10.6) and 21.7 months (95% CI 8.8–38.4), respectively [29].

In the multicenter Phase III TROPiCs-02 study, patients with HR+ HER2− advanced breast cancer were randomized 1:1 on SG versus physicians’ choice of chemotherapy. Patients were required to have progressed on a CDK4/6i and at least 2 chemotherapy agents, including a taxane. The study met its primary endpoint with statistically significant improvement in PFS on SG of 5.5 months versus 4.0 months on physicians’ choice of chemotherapy (HR 0.66; 95% CI, 0.53–0.83; *p* = 0.0003). Patients also had an improved ORR at 21% versus 14% on SG, as well as an improved CBR of 34% versus 22% [31]. At the second interim analysis, an OS benefit of 3.2 months was seen (HR = 0.79, CI 0.65–0.96, *p* = 0.02). Patients on SG had an OS of 14.4 months compared to 11.2 months for patients on the treatment of their physician’s choice [30]. SG provided an OS benefit in patients who had received at least three prior therapies, making treatment an important option in pretreated patients. This antibody drug conjugate remains an option for patients with advanced HR+, HER2− breast cancer, particularly when transitioning to chemotherapy is a consideration.

### 2.7. Continuing CDK4/6i

Continuing a CDK4/6i at time of progression has been hypothesized as potentially effective, especially in patients treated with palbociclib first-line therapy. In the Phase II MAINTAIN clinical trial, 119 patients who had progressed on CDK4/6i and endocrine therapy were randomized to fulvestrant or exemestane with or without ribociclib 1:1. Eighty-four percent of patients prior to randomization were on palbociclib. Patients randomized to ribociclib had a mPFS of 5.29 versus 2.76 months in the control arm (HR 0.59, 95% CI 0.39–0.95, *p* = 0.006). Of note, 25% of patients remained on this therapy at 12 months versus 7% in the control arm of fulvestrant or exemestane [33]. In a subset analysis, the PFS benefit of ribociclib in the second line was statistically significant in patients who had palbociclib as first-line therapy (*n* = 103, HR 0.58 95% CI 0.38–0.90) but not in patients who had received ribociclib in the first line (*n* = 14, HR 0.50 95% CI 0.15–1.70).

In the Phase II trial PACE, patients who were on a CDK4/6i first-line therapy for at least 6 months were randomized to fulvestrant vs. fulvestrant and palbociclib vs. fulvestrant, palbociclib, and avelumab [42]. There was no significant difference between the mPFS of fulvestrant (4.8 months) and fulvestrant and palbociclib (4.6 months) in the 166 patients enrolled. Interestingly, in patients with PIK3CA-mutated tumors, continuing palbociclib with or without immunotherapy conferred an improved PFS (HR 0.56 90% CI 0.23–0.99), suggesting that molecular targets may allow us to better target patient populations who would benefit from continuing CKD4/6i.

Abemaciclib post-progression was studied in a retrospective, multicenter analysis. Patients who had progressed on ribociclib or palbociclib and were then treated with abemaciclib had an mPFS of 5.3 months with a median OS of 17.2 months [43]. Patients were heavily pretreated and were on a median of 5 lines of therapy prior to abemaciclib, with a median 3 lines of endocrine therapy. Interestingly, the mPFS was significantly longer in patients who received CDK4/6i sequentially at 8.4 months, rather than non-sequentially at 3.9 months (*p* = 0.0013, 95% CI 2.9–5.7 months). Most patients received abemaciclib with an anti-estrogen agent. A total of 47% of subjects received abemaciclib with fulvestrant, 27.6% received it with an aromatase inhibitor, and 19.5% of patients received it as a monotherapy. Thirty six percent of patients received abemaciclib for ≥6 months. RB1, ERBB2, and CCNE1 alterations were noted in patients who were not responsive to abemaciclib, suggesting that tumor biomarkers may help in the selection of patients who will respond to a rechallenge to a CDK4/6i.

While MAINTAIN and PACE outcomes are conflicting, data from PALOMA-2 and PACE suggest that palbociclib may have different activity than other CDK4/6i. It may be reasonable to continue CDK4/6i at the time of progression, especially in patients treated with palbociclib in the first-line setting; however, additional data are needed to confirm this as a treatment strategy. Further understanding of tumor mutations and mechanisms of resistance to endocrine resistance may allow us to target patients who would benefit from continuing CDK4/6i.

## 3. Targeting the Estrogen Receptor

### 3.1. Oral Selective Estrogen Receptor Degraders

Oral selective estrogen receptor degraders (SERDs) belong to a promising class of drugs given their ability to overcome ESR1 mutations, oral delivery, and improved bioavailability. Selective estrogen receptor modulators (SERMs) such as tamoxifen directly target ER-alpha and behave as mixed agonist and antagonists towards the receptor. Endocrine resistance can occur through ER reactivation caused by acquired ESR1 mutations. SERDs bypass this mechanism of resistance by binding to the estrogen receptor, creating an unstable complex, thereby inducing degradation through proteosomes. The FDA has approved the oral SERD elacestrant for patients with HR+, HER− mBC with ESR 1-mutated disease with progression on at least one endocrine therapy. Phase I/II trials for other SERDs have explored their safety and efficacy at various doses, which have been well tolerated with few Grade 3 or 4 events [44,45,46,47] (Table 2). Oral SERDs are undergoing investigation as a monotherapy and in combination with CDK4/6i and other drugs such as everolimus (Table 3).

In the Phase III EMERALD study, patients with HR+, HER− mBC who progressed on CDK4/6i therapy were randomized to monotherapy elacestrant (Radius) versus the investigator’s choice of monotherapy endocrine therapy [39]. Patients were eligible if they had progression of disease on least one line of prior endocrine therapy and no more than one prior line of chemotherapy. Patients had to have progressed on a CDK4/6i, and 29% of patients had prior exposure to fulvestrant. Eighty percent of patients never had chemotherapy. A total of 92% of patients had adverse events (AE) in the elacestrant arm (*n* = 237), with 27% of adverse events being Grade 3 or 4. Grade 3 or 4 events were limited to nausea (2.5%), fatigue (0.8%), vomiting (0.8%), decreased appetite (0.8%), arthralgias (0.8%), back pain (2.5%), increased AST (1.7%), and increased ALT (2.1%) (Table 2).

Elacestrant had a superior PFS rate in comparison to fulvestrant, and the hazard ratio improved in the subgroup of patients with ESR1-mutated tumors (Table 3). The mPFS in the ITT population on elacestrant was 2.8 months, in comparison to 1.9 months for fulvestrant monotherapy (49). Twelve-month PFS rates on elacestrant were 22.3% compared to 10.2% of patients who received fulvestrant (HR 0.68, 95% CI 0.52–0.90). ESR1 mutations were present in 47.8% of patients’ tumor samples, and in that subgroup, mPFS was 3.8 months compared to 1.9 months for those on fulvestrant monotherapy. Twelve-month PFS rates on elacestrant in patients with ESR1-mutated tumors were 26.8% compared to 8.4% on fulvestrant, with an improved HR of 0.50 (95% CI 0.34–0.74).

Elacestrant was associated with an improved PFS in patients with endocrine-sensitive disease. In patients who were on a CDK4/6i for at least 12 months, elacestrant was associated with an improved PFS of 3.8 months vs. 1.9 months on the standard of care arm (HR 0.6, CI 95% 0.45–0.83) [53]. In patients who were on CDK4/6i for at least 18 months, elacestrant was associated with an improved PFS of 5.45 months vs. 3.29 months on the standard of care arm (HR 0.7, CI 95% 0.48–1.02) [53]. Elacestrant is currently FDA approved for patients with ESR1 mutations who have progressed on one line of endocrine therapy, and it remains an important option for patients, especially in those that have endocrine sensitive disease.

Camizestrant (AstraZeneca) was studied in the Phase II SERENA-2 trial. Patients were eligible if they had received ≥ 1 line of endocrine therapy and no more than 1 prior chemotherapy regimen. Patients were randomized in a 1:1:1 fashion to camizestrant 75 mg, camizestrant 150 mg, or fulvestrant. A total of 50% of patients had a prior CDK4/6i, 42% of patients had prior SERM, and 37% of patients had an ESR1 mutation. At a median follow up of 16 months, camizestrant 75 mg was associated with a 7.2 month mPFS compared to 3.7 months (HR 0.58, 90% CI 0.41–0.81). Camizestrant 150 mg was associated with an mPFS of 7.7 months (HR 0.67 90% CI 0.48–0.92) [49]. In subgroup analysis, patients with ESR1-mutated disease had statistically significantly improved PFS over ESR1-undetectable tumors. Patients with ER-driven disease, defined as patients who had received CKD4/6i and endocrine therapy for at least 12 months, also had statistically significant improved PFS over fulvestrant monotherapy. These findings suggest EMERALD’s findings perhaps are a class effect and that patients with somatic ESR1 mutations and endocrine-sensitive disease may respond better to oral SERDs. This is further supported by Phase I SERENA-1 findings, where camizestrant and palbociclib (*n* = 25) had a CBR of 28% in a heavily pretreated population, suggesting limited benefit in patients with endocrine-resistant disease [52].

Camizestrant has a unique AE profile. At the 150 mg dosing of camizestrant, 24.7% of patients (18/73) had photopsia; however, none of these events were Grade 3 or higher. Other AEs present in more than 10% of the study population included sinus bradycardia, fatigue, anemia, asthenia, and arthralgias (Table 2). Grade 3 events or higher were limited to 2.7% (2/73) of the study population, and there were no Grade 5 events.

Imlunestrant (Lilly) was studied as a monotherapy in the Phase I trial EMBER. Patients were eligible if they had up to 3 prior therapies for metastatic disease in the dose escalation period. A total of 92% of patients had progressed on a CDK4/6i. The CBR was 40.4% (42/104), and the ORR was 8.0% (6/75) [48]. Rintodestrant (G1 Therapeutics), another oral SERD, had a limited CBR of 28% despite limited prior exposure to CDK4/6i (69%) in a Phase I trial (*n* = 67) [47]. These two oral SERDs had an AE profile similar to elacestrant, with predominantly GI and musculoskeletal toxicities noted (Table 2).

Other oral SERDs have shown similar CBRs in Phase I studies but have failed to show a PFS benefit at Phase II interim analysis. Amcenestrant (Sanofi) monotherapy in the Phase I AMEERA-1 had a CBR of 28.3%; however, the Phase II AMEERA-3 and Phase III AMEERA-5 trials were suspended due to a lack of clinical benefit at interim analysis [50,54]. Giredestrant (Roche) monotherapy in a Phase I trial had an impressive PFS of 7.2 months and a CBR of 48%; however, the monotherapy Phase II alcelERA trial failed to meet the primary endpoint of improving PFS as a monotherapy in comparison to endocrine monotherapy of the physician’s choice and has since been suspended [44,55].

Oral SERDs are a promising line of therapy for patients with ESR1-mutated, endocrine-sensitive tumors. The FDA approval of elacestrant in patients with ESR-mutated tumors who have progressed on at least one line of endocrine therapy is an important addition to treatment options available for HR+, HER− mBC. Additional data are needed from other oral SERDs; however, there have been promising updates recently suggesting alternative oral SERDs may be available in the future. Variations in ORR between camizestrant and imlunestrant may be due to differences in prior CDK4/6i exposure. Additional prospective data are needed to determine its role in future therapy sequencing.

### 3.2. Proteolysis-Targeting Chimeras (PROTAC)

Proteolysis-targeting chimeras (PROTAC) are under investigation as a possible treatment for HR+, HER2− mBC (Table 4). PROTACs are hetero-bifunctional molecules composed of two active domains and a linker. ARV-471 recruits E3 ligase, which degrades ubiquitinated proteins into small peptides. This active degradation may result in the improved knockdown of tumor ER relative to fulvestrant. In vitro studies demonstrated that ARV-471 can achieve potent (<1 nM) and robust (>90%) ER degradation even in the presence of the clinically relevant ESR1 mutations Y537S and D538G. In vivo ARV-471 inhibits tumor growth in multiple mouse xenograft models. At low doses, ≥85% tumor growth inhibition was noted in preclinical studies [56].

In the Phase I/II VERITAC trial, patients received monotherapy oral PROTAC ARV-471. Patients were required to have at least 1 prior line of endocrine therapy, prior treatment with a CDK4/6i, and could not receive more than 1 line of chemotherapy. Patients had a median of 4 lines of prior therapy. A total of 100% of patients had received a CDK4/6i, and 79% of patients had prior treatment with fulvestrant. The CBR was 38%, with an mPFS of 3.7 months in the IIT population, and CBR was 51% with an mPFS of 5.7 months in patients with ESR1-mutated tumors. ARV-471 had 15 cases of Grade 3/4 events our of 71 patients (21%), limited to QT prolongation, thrombocytopenia, hyperbilirubinemia, fatigue, neutropenia, and decreased appetite [42]. ARV-471 shows promising ER degradation and CBR in a heavily pretreated population. Additional clinical data are needed to determine if this treatment can be incorporated into the mBC standard of care and is being investigated in comparison to fulvestrant in VERITAC 2.

### 3.3. Selective Estrogen Receptor Covalent Antagonists

Selective estrogen receptor covalent antagonists (SERCA) are under investigation as a treatment for HR+, HER− mBC. SERCAs are small molecules that are covalent irreversible antagonists of ER-alpha. They induce confirmational changes of the estrogen receptor without degrading it. The SERCA small-molecule H3B-6545 demonstrated single agent activity in CDK4/6i-resistant xenograft models, in which fulvestrant failed to demonstrate anti-tumor activity [65].

In a Phase I/II trial of SERCA H3B-6545, patients with estrogen receptor-positive disease that had progressed through at least one line of therapy had a CBR of 40.3% and ORR of 16.7%. In this trial, 85% of patients had received a prior CDK4/6i. The median duration of response was 7.6 months, with a median PFS of 5.1 months. Thirty five percent of patients enrolled in this trial had an ESR1 mutation. The most common AEs were GI toxicity, anemia, fatigue, and Grade 1 or 2 sinus bradycardia [58,59]. An escalation study if H3B-6545 with palbociclib is ongoing in patients with HR+, HER− mBC with up to one prior chemotherapy and one prior CDK4/6i treatment. At preliminary analysis, the combination was tolerable with no major interactions through PK analysis [66]. H3B-6545 shows promising response rates; however, additional clinical data are needed to determine clinical benefit.

### 3.4. Complete Estrogen Receptor Antagonists (CERANs)

Complete estrogen receptor antagonists (CERANs) have been studied in preclinical settings and in a Phase I trial. Its mechanism of action is based on complete estrogen receptor deactivation. The estrogen receptor has two transcription activation functions: AF1 and AF2. SERDs and SERMs primarily deactivate AF2. CERANs antagonize AF1 and recruit N-CoR, which inactivates AF2, therefore completely inhibiting ER functioning [67]

In a Phase IA trial of 27 patients with HR+, HER2− mBC patients who were pretreated including prior CDK4/6i were offered CERAN OP-1250. Three Grade 3 or higher AEs were noted, and these consisted of neutropenia, nausea, and emesis. The ORR was 9% (2/23), with a CBR of 21% (4/19) including patients with an ESR1 mutation. Two PR were noted, and both harbored an ESR1 mutation [60]. A Phase II portion of this study is expected to continue as a monotherapy and in combination with a CDK4/6i. OP-1250 will be moving forward with a Phase IA/II study.

### 3.5. Selective Estrogen Receptor Modulators (SERMs)

Lasofoxifene is a selective estrogen receptor modulator (SERM) with improved bioavailability and volume distribution in comparison to other SERMs. It has been shown in mouse xenograft models to show improved efficacy against ESR1-mutated tumors in comparison to fulvestrant. Ligand binding assays showed that lasofoxifene had superior affinity to ESR1-mutated tumors in comparison to fulvestrant. It was also shown to reduce tumor growth in comparison to fulvestrant [68].

The open-label, Phase II ELAINE 1 trial evaluated the SERM lasofoxifene 5 mg versus fulvestrant in patients with somatic ESR1-mutated HR+ HER2− mBC. One hundred and three patients were randomized. Patients were required to have tolerated an aromatase inhibitor and CDK4/6i for at least 12 months. Lasofoxifene had a numerically greater PFS of 6.04 months (95% CI, 2.82–8.04) compared to fulvestrant with its median PFS of 4.04 months (95% CI, 2.93–6.04), *p* = 0.138 (HR, 0.699 (95% CI, 0.445–1.125)). PFS at 12 months was 30.7% on lasofoxifene versus 14.1% on fulvestrant. CBR and ORR were numerically superior on lasofoxifene; however, this was not statistically significant. On lasofoxifene, CBR was 36.5% versus 21.6% on fulvestrant (*p* = 0.12). The objective response rate on lasofoxifene was 13.2% versus 2.9% on fulvestrant (*p* = 0.12) [69].

In this Phase II clinical trial, lasofoxifene was not inferior to fulvestrant in tumors with ESR1-mutated mBC. Lasofoxifene when used at lower doses for osteoporosis (0.5 mg) had similar adverse events to other SERMS, such as thromboembolism, endometrial hypertrophy, and arthralgias [70]. Lasofoxifene may be a tolerable option for patients with ESR1-mutated mBC; however, additional data are needed in comparison to available endocrine therapies.

## 4. Overcoming Endocrine Resistance

### 4.1. Immunotherapy

Pembrolizumab, a PD-1 inhibitor that blocks inhibitory signaling on T cells, currently is the only immunotherapy with FDA approval for use in patients with breast cancer. It currently has two indications for use in patients with triple-negative breast cancer, which encompasses subsets of patients with localized and metastatic disease. Given the relatively low number of tumor-infiltrating lymphocytes (TILs) found in HR-positive disease relative to other breast cancer subtypes, it has been hypothesized that immunotherapy would be less effective in patients with HR-positive disease. This has largely been correct; overall response rates to single-agent immunotherapy in the metastatic setting range from 3 to 12% [71,72]. In ISPY2, the addition of neoadjuvant pembrolizumab to patients with ER-positive disease and a high-risk Mammaprint improved pathologic complete response rates (pCR), which suggests that some ER-positive diseases with higher-risk features may be more immunogenic. Total pCR rates, however, remained modest, with 14.8% in the control arm and 28.0% in the pembrolizumab arm achieving a pCR [73].

Despite endocrine therapy causing an overall decline in TILs, in vivo studies have shown a synergistic effect of CDK4/6i and immunotherapy. Prospective trials combining immunotherapy and CKD4/6i have unfortunately had mixed results due to poor tolerability. In a Phase I/II trial of HR+, HER− mBC, 23 patients received palbociclib, pembrolizumab, and letrozole. Sixteen patients received this therapy as first-line treatment. Eighty seven percent of patients had Grade 3 toxicities (20/23), and thirty percent of patients had Grade 4 toxicity (7/23). Grade 3/4 events included cytopenias (most commonly neutropenia), nausea, vomiting, pneumonitis, infection, transaminase, and bowel perforation. The mPFS was 25.2 months (95% CI 5.3—not reached), which is comparable to PALOMA-2’s mPFS of 24.8 months. The median OS was 36.9 months (95% CI 36.9—not reached). A total of 31% (5/16) of patients in the newly diagnosed cohort achieved complete response (CR), 25% (4/16) had a partial response (PR), and 31% (5/16) had stable disease (SD). While response rates are impressive, the toxicities noted in this trial would make this regimen challenging as a standard of care [74]. In the Phase II CheckMate 7A8 trial, patients were randomized to receive neoadjuvant palbociclib and anastrozole with or without nivolumab. This trial halted enrollment due to increases in hepatic and lung toxicity. Grade 3 transaminase was noted in 29% of patients (*n* = 6). Overall, 5% of patients (*n* = 1) had Grade 2 immune0mediated lung disease, and 5% (*n* = 1) had pneumonitis [75].

The Phase II PACE trial also had numerically higher percentages of Grade 3 and 4 adverse events in the palbociclib, fulvestrant, and avelumab arm in comparison to the palbociclib and fulvestrant arm. However, in patients with specific somatic mutations, an improved PFS was noted with the addition of immunotherapy, suggesting that offering this therapy to targeted populations perhaps at variable dosing could be of benefit in the future.

### 4.2. Capivasertib and Fulvestrant

Capivasertib is a selective inhibitor of AKT. The AKT pathway is downstream of PTEN and PIK3CA and upstream of mTOR. AKT pathway activation can occur with or without genetic alteration and is associated with endocrine resistance. In the Phase II FAKTION trial, patients were randomized to capivasertib and fulvestrant vs. fulvestrant monotherapy, and the investigational agent significantly improved PFS and OS in postmenopausal HR+/HER− mBC. Patients had not received prior CDK4/6i, as the trial was designed prior to the widespread use of these drugs. In a subset analysis, the benefit of capivasertib was limited to patients with tumors that had AKT pathway-mutated tumors [76].

In the Phase III CAPItello -291 trial, patients were randomized 1:1 to capivasertib and fulvestrant vs. placebo and fulvestrant. Patients were allowed no more than 2 lines of prior endocrine therapy, no more than 1 line of chemotherapy. Patients were also allowed on trial with a HbA1c of less than 8.0%. Of trial participants, 69% had prior CDK4/6i, and 40% had AKT pathway alterations noted. In the investigational arm at two years, capivasertib and fulvestrant had an improved investigator-assessed PFS of 7.2 months compared to 3.6 months in the placebo arm (HR 0.6 95% CI 0.51–0.71, *p* < 0.001). In patients with tumors that expressed AKT pathway alterations, this PFS benefit was more pronounced at 7.3 months compared to 3.1 months in the placebo arm (HR 0.5, 95% CI 0.38–0.65 *p* <0.001). In subgroup analysis, HR remained statistically significant in patients that had prior CDK4/6i. Gastrointestinal side effects were common, with 72.4% of patients experiencing diarrhea of any grade and 9.3% of patients suffered Grade 3 diarrhea [57]. Rash was also noted in 22.0% of patients, and 5.4% of patients had Grade 3 rash (Table 4). Despite targeting a downstream pathway of PIK3CA and allowing patients with diabetes on trial, hyperglycemia was only noted in 16.3% of patients in the study, and only 2.3 of patients had Grade 3 events. Phase Ib CAPItello-292 is ongoing comparing the use of capivasertib, palbociclib, and fulvestrant to palbociclib and fulvestrant in patients with prior endocrine therapy, with plans to expand to Phase III with prior CDK4/6i use restrictions. Data available suggest that this targeted therapy may provide benefit in patients with HR+ HER− mBC.

### 4.3. Samuraciclib plus Fulvestrant

Samuraciclib is a selective inhibitor of CDK7, which, when inhibited, blocks the necessary phosphorylation required for cell cycle progression and the transcription of oncogenic and anti-apoptotic genes. In xenograft models, samuraciclib was most active in suppressing tumor volume in combination with fulvestrant in comparison to treatment as a monotherapy.

In a Phase I/II multicenter study, patients with HR+ HER2− mBC who had previously been treated with a CDK46i were started on samuracuclib. The first six patients were started on samuraciclib 240mg daily with fulvestrant, and patients were then escalated to samuraciclib 360 mg daily. The overall CBR of patients including both dose exposures was 36% at 24 weeks. Seventy two percent of patients had tumor shrinkage noted. Patients whose tumors contained a p53 mutation (*n* = 6) had a median PFS of 7.9 weeks, whereas the p53 wild type (*n* = 19) had a median PFS of 32 weeks (Table 4). Baseline tumor p53 wild-type status may predict benefit from samuraciclib, which can be obtained by ctDNA [61]. Preclinical data suggests that CDK7 inhibition activates the p53 pathway, inducing apoptosis. This mechanism of action may explain why patients with p53-mutated disease have poor PFS responses. Samuraciclib is being fast tracked by the FDA, and additional data are needed to further characterize its benefit.

### 4.4. Selective Androgen Receptor Modulators

Enobosarm is a selective androgen receptor modulator (SARM). The androgen receptor (AR) is present in up to 95% of HR+ breast cancer patients and is hypothesized to have tumor suppressor qualities in patients with HR+ mBC [77]. AR agonism has suppressed the growth of HR+ endocrine-sensitive and -resistant breast cancers in preclinical models. In a cell model, palbociclib and enobosarm showed a decreased proliferative rate with combination therapy than with either agent alone. No AR biomarker was used in this study [78].

A Phase II open-label study with 136 patients evaluated the efficacy and safety of enobosarm in heavily pretreated women with AR+/HR+ mBC (Table 4). Patients must have progressed on prior endocrine therapy. The median PFS was 5.6 months and 4.2 months in the 9 mg and 18 mg groups, respectively. The most common AEs reported were improvements in mobility, anxiety/depression, and pain [63]. AR staining was predictive of response, with CBR for AR ≥ 40% of 80%, and < 40% was 18% (*p* < 0.0001). The best objective tumor response in patients with ≥ 40% AR was 48%, and < 40% is 0% (*p* < 0.0001) [62]. Enobosarm is a well-tolerated targeted therapy being offered through Phase III clinical trials. Additional prospective data are needed to further determine its place in therapy sequencing.

### 4.5. Fibroblast Growth Factor Receptor (FGFR) Inhibitors

Fibroblast growth factor receptor (FGFR) pathway overexpression or hyperexpression is associated with CDK4/6i resistance and shortened PFS. CDK4/6i indirectly trigger the dephosphorylation of retinoblastoma tumor suppressor protein by blocking CDK4/6, thereby blocking the cell cycle transition from the G1 to the S phase. FGFR inhibitors have been hypothesized to help overcome CDK4/6i resistance and studied as a monotherapy and in conjunction with CDK4/6i.

Erdafitinib is a selective FGFR inhibitor, which was studied in a Phase II, open-label, single arm study from NCI-MATCH. In this study, 48 patients with FGFR amplifications or mutations, including 13 with breast cancer, received FGFR inhibitor monotherapy. The median PFS was 3.4 months, and the 6-month PFS rate was 15% (90% CI, 8–31%). For patients with tumors harboring FGFR fusions, the response rate was 22% (90% CI, 4.1–55%), and the 6-month PFS rate was 56% (90% CI, 31–100%). The study did not meet its 16% ORR threshold across all FGFR alterations, possibly due to the heterogeneity of these alterations (Table 4) [64].

In a Phase Ib trial of 18 patients combining the FGFR pan-inhibitor erdafitinib with fulvestrant and palbociclib, the median PFS was 3 months, and the clinical benefit rate at 6 months was 28%. Four patients had not completed treatment at the time of analysis, and all patients had prior CDK4/6i treatment. Higher PFS was noted in 6/8 patients with high levels of FGFR1 amplification and in both patients with FGFR3 amplification [79]. This class of drugs is currently undergoing further clinical investigation and may be of benefit to patients with specific FGFR alterations.

## 5. Discussion

This paper reviews treatment options for patients with HR+, HER2− mBC post-front-line CDK4/6i with endocrine therapy. When considering second-line therapy, patients’ overall health and tumor biology should be considered. This paper posits a treatment algorithm when considering second-line therapy options (Figure 1).

PARPi are currently FDA approved for first through third line therapy for patients with mBC. Preclinical data and retrospective analysis suggest that patients with gBRCA mutations have shorter response rates to CDK4/6i in comparison to non-mutated patients, suggesting that gBRCA may confer endocrine resistance. Given the lack of OS benefit with PARPi use, we consider this therapy particularly in patients with relatively low burdens of disease and good performance status. In the second line, patients should also have other endocrine-based and/or targeted therapies considered. For patients with PIK3CA-mutated tumors, alpelisib with fulvestrant should be considered as a second-line therapy, as this therapy option has reported a PFS of 5.6–7.3 months in prospective studies [21,22]. Consideration of alpelisib’s side effects, particularly hyperglycemia, should be discussed with patients prior to treatment. For patients with ESR1-mutated tumors, elacestrant should be considered second line, particularly in patients with endocrine-sensitive disease. For tumors without targeted mutational options, second-line endocrine therapy with or without everolimus should be considered. In patients with prior endocrine therapy exposure, monotherapy fulvestrant has been associated with a limited PFS of 1.9 months [24,39]. However, this monotherapy option remains well tolerated, and is an attractive option for a select patient population.

Sacituzumab govitecan and trastuzumab deruxtecan are antibody drug conjugates with a noted PFS and OS benefit post-CDK4/6i. In TROPiCs-02, sacituzumab govitecan improved mPFS by 1.5 months and mOS by 3.2 months compared to the control arm [30,31]. In DESTINY-Breast04, trastuzumab deruxtecan improved mPFS by 4.7 months, and improved mOS by 6.4 months over the control arm [32]. In these trials, patients were required to have progressed on prior endocrine therapy and chemotherapy. These antibody drug conjugates are well tolerated and could be considered prior to chemotherapy given the OS benefit in select patients, particularly those with rapidly progressing disease that may be unable to tolerate chemotherapy. Given the pronounced overall survival benefit noted in DESTINY-04, these authors recommend the use of T-DXd prior to sacituzumab govitecan. Patient co-morbidities should be used to guide decision making, particularly in the presence of underlying lung or cardiac disease.

Novel targets of the estrogen receptor that are currently under clinical investigation were also reviewed. Additional oral SERDs are under investigation with promising data, particularly in patient populations with ESR1-mutated tumors. Most oral SERDs under investigation are associated with musculoskeletal or GI side effects, with the noticeable exception of camizestrant, which was associated with photopsia and bradycardia. Other estrogen receptor-targeted therapies have CBRs and ORRs similar to oral SERDs and have tolerable AE profiles, which were reviewed above.

Clinical investigations of therapy options that attempt to overcome endocrine resistance are ongoing. Preclinical studies suggested that anti-PDL1 therapy promoted synergistic tumor regression in combination with CDK4/6i; however, clinical data with this combination have shown an untenable side effect profile or haves not been statistically significant. In the PACE trial, continuing CDK4/6i with fulvestrant and immunotherapy numerically improved PFS and OS; however, this difference was not statistically significant in the IIT population. The PFS benefit was significant in patients with PIK3CA-mutated tumors, suggesting a pathway to further personalize treatment to tumor biology. Other therapies that attempt to overcome endocrine resistance, with or without CDK4/6i, have shown promising CBRs and ORR. Additional prospective clinical data are needed to determine these therapies’ efficacies.

## 6. Conclusions

The first-line treatment for HR+ HER2− mBC remains CDK4/6i with endocrine therapy. Optimal sequencing post-CDK4/6i should be personalized to the patient’s functional status and tumor biology (Figure 1). Targeted or additional endocrine therapy should be considered prior to the use of chemotherapy. The oral SERD elacestrant was recently FDA-approved in patients with somatic ESR1 mutations that have progressed on a prior line of endocrine therapy. Increasing PFS benefit was seen to correlate to the degree of overall endocrine sensitivity. T-DXd and sacituzumab govitecan have shown impressive OS benefits of 6.4 months and 3.2 months, respectively, over the investigators’ choice of therapy. In prospective trials showing this benefit, patients had to have prior chemotherapy exposure. Earlier use of T-DXd in patients with HER2 low disease should be considered prior to chemotherapy in patients with a high burden of disease, as T-DXd has been associated with an impressive response rate and OS benefit. T-DXd use prior to chemotherapy is under active investigation in DESTINY-06.

Additional oral SERDs are under investigation as a monotherapy and in conjunction with other therapies, such as continued CDK4/6i or everolimus. This paper has reviewed adverse events and response rates associated with specific oral SERDs, which can be considered when reviewing available clinical trials. Other targeted therapies that overcome endocrine resistance or directly target the estrogen receptor have promising efficacy post-CDK4/6i and should be considered when available through clinical trials.

## Figures and Tables

**Figure 1 cancers-15-01855-f001:**
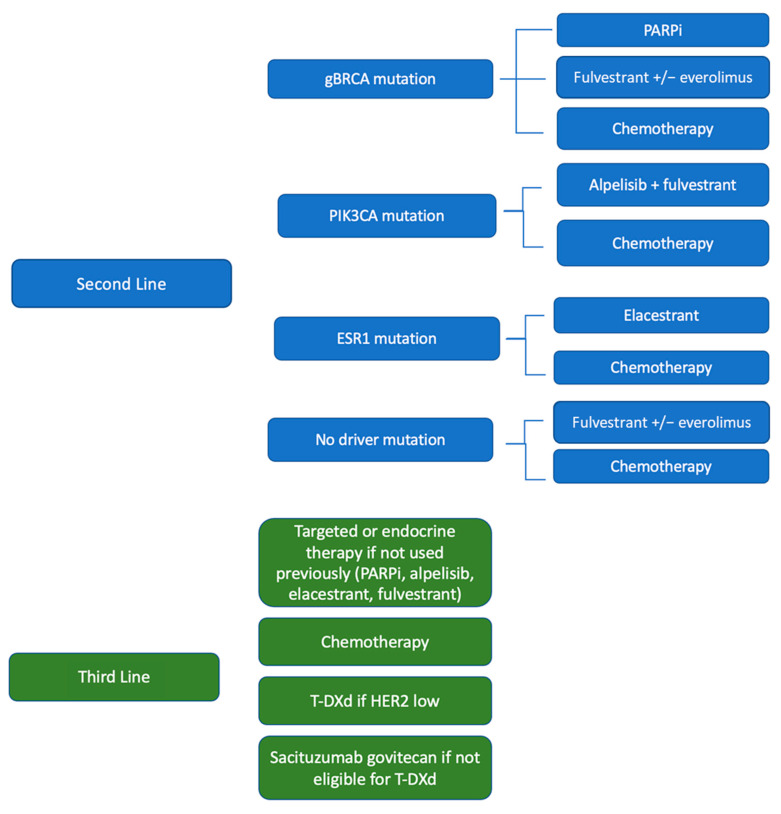
Suggested post-first-line CDK4/6i sequencing of therapy. T-DXd = trastuzumab deruxtecan, gBRCA = germline BRCA mutation, PARPi = PARP inhibitor. Please note that there are no data on the use of fulvestrant post-elacestrant.

**Table 1 cancers-15-01855-t001:** Post-CDK4/6i therapy options in HR+ mBC.

Drug	Trial	*n*	Prior Lines of Therapy	Prior CDK4/6i	PFS	Response Rate	Reference	
**Alpelisib fulvestrant**	Prospective Phase II, BYLieve trial, Cohort A	121	Must have had CDK4/6i Patients had not received more than 3 lines of therapy	99%	7.3 months	ORR 17% CBR 55%	[21]	
	Prospective Phase II, BYLieve trial, Cohort C	112	Patients had not received more than 3 lines of therapy	84%	5.6 months	ORR 28% CBR 43%	[22]	
**Fulvestrant**	Prospective, Phase III EMERALD trial	165	Must have had CDK4/6i ≤2 prior lines of endocrine therapy ≤1 line of chemotherapy	100%	1.9 months	6-month PFS rate 23% 12-month PFS rate 10%	[23]	
	Prospective, Phase II, VERONICA trial	51	Must have had CDK4/6i ≤2 prior lines of endocrine therapy No prior cytotoxic chemotherapy	100%	1.94 months	CBR 13.7%	[24]	
**Exemestane and everolimus**	Retrospective	41	Must have had CDK4/6i Median 4 prior lines of therapy	100%	4.2 months	ORR 17.1% CBR 17.1%	[25]	
	Retrospective	20	Must have had CDK4/6i	100%	5.8 months	NA	[26]	
	Retrospective	17	Must have had CDK4/6i	100%	5.7 months	6-month PFS 18%	[27]	
	Retrospective	12	Prior therapy CDK4/6i	100%	11.7 months	NA	[28]	
**Sacituzumab govitecan**	Prospective, Phase I/II	32	Progressed on at least one prior endocrine therapy and chemotherapy Subgroup analysis in pts with prior CDK4/6i	100%	3.8 months	ORR 8% CBR 12%	[29]	
	Prospective, Phase III TROPiCs-02	272	Progressed on a CDK4/6i and at least 2 chemotherapy agents including a taxane	98%	5.5 months	ORR at 21% Improvement PFS compared to control: 1.5 months Improvement in OS compared to control: 3.2 months	[30,31]	
**Trastuzumab deruxtecan**	Prospective, Phase III DESTINY-Breast04 *	331	Must have received at least one line of chemotherapy Must have received at least one line of endocrine therapy	70.4%	10.1 months	CBR: 71.2%, ORR 52.6% Improvement PFS compared to control: 4.7 months Improvement in OS compared to control: 6.4 months	[32]	
**Continuing CDK4/6i**	Prospective, Phase II MAINTAIN	120	Progressed on CDK 4/6i and ET	97%	5.29 months	12-month PF rate 25%	[33]	
	Prospective Phase II PACE	220	Progressed on CDK 4/6i and ET	99.5%	Fulvestrant: 4.8 months Fulvestrant and palbociclib: 4.6 monthsFulvestrant, palbociclib, avelumab: 8.1 months	CBR fulvestrant monotherapy: 29.1% CBR fulvestrant, palbociclib: 32.4% CBR fulvestrant, palbociclib, avelumab: 35.2%	[34]	

ORR = overall response rate, CBR = clinical benefit rate, PFS = progression-free survival, PF = progression-free, OS = overall survival. * = HER2 low, HR+ mBC patients only.

**Table 2 cancers-15-01855-t002:** Adverse events of oral SERDs under investigation.

	Elacestrant (Radius)	Imlunestrant (Lilly)	Camizestrant (AstraZeneca)	Rintodestrant (G1 Therapeutics)
**Phase**	III	I	II *	I
** *n* **	237	141	73	67
**Total % Adverse Events**	92%	92%	90.4	70%
**Adverse Events ≥ 10% (%)**	Nausea (35%) Fatigue (19%) Vomiting (19%) Decreased appetite (14.8%) Arthralgia (14.3%) Diarrhea (13.9%) Back pain (13.9%) AST increased (13.1%) Constipation (12.2%) Headache (12.2%) Hot flashes (11.4%) Dyspepsia (10.1%) ALT increased (9.3%)	Nausea (37%) Diarrhea (28) Fatigue (27%) Arthralgias (16%) Headache (12%) Cough (11%) Hot flush (11%) UTI (11%) Anemia (11%)	Photopsia (24.7%) Bradycardia (26%) Fatigue (17.8%) Anemia (15.1%) Asthenia (15.1%) Arthralgias (12.3%)	Hot flush (24%) Fatigue (21%) Nausea (19%) Diarrhea (18%) Vomiting (10%)
**Grade ≥ 3 AE (%)**	Nausea (2.5%) Back pain (2.5%) ALT increased (2.1%) AST increased (1.7%) Headache (1.7%) Fatigue (0.8%) Vomiting (0.8%) Decreased appetite (0.8%) Arthralgia (0.8%)	Nausea (1%) Diarrhea (1) Fatigue (1%) Anemia (1%)	Fatigue (1.4%) Anemia (1.4%) Arthralgia (1.4%) Pain in any extremity (1.4%) Hyponatremia (1.4%) Increased blood pressure (1.4%)	Increased ALT (1%) Cerebral hemorrhage (1%)
**References**	[39]	[48]	[49]	[47]

AE = adverse events. * Camizestrant 150 mg dosing was used in reporting AE.

**Table 3 cancers-15-01855-t003:** Oral SERD trials in the metastatic setting.

Drugs Under Investigation	Monotherapy Trials	Prior CDK4/6i	CDK4/6i Combination Trials	Everolimus Combination Trials	Monotherapy Median PFS	Monotherapy Response Rate	References
**Elacestrant** **(Radius)**	Phase III EMERALD	100%	Phase II (NCT04791384) CDK4/6i = abemaciclib	NA	Phase I—4.5 months Phase III EMERALD—2.8 months	Phase I CBR—42.6%, ORR 19.4% Phase III EMERALD – 12-month PFS rate 22.3% – PFS 2.8 months, improvement over fulvestrant 0.9 months	[39,50]
**Giredestrant** **(Roche)**	Phase II acelERA—trial suspended due to lack of PFS benefit at interim analysis	NA	Phase III persevERA (NCT04546009) CDK4/6i = palbociclib	Phase III evERA (NCT05306340)	Phase I—7.2 months Phase II acelERA—trial suspended due to lack of PFS benefit over physician’s choice of ET	Phase I CBR– 48%	[44]
**Amcenestrant** **(Sanofi)**	Phase II AMEERA-3—trial suspended due to lack of PFS benefit at interim analysis	NA	Phase III AMEERA-5—trial suspended due to lack of clinical benefit at interim analysis	NA	NA	Phase I CBR—28.3%, ORR 10.9% Phase II AMEERA-3—trial suspended due to lack of PFS benefit at interim analysis	[50]
**Imlunestrant** **(Lilly)**	Phase I EMBER (NCT04188548)	92%	Phase 3 EMBER-3 (NCT04975308) CDK4/6i = abemaciclib	Phase I EMBER (NCT04188548)	Phase I EMBER, post-CDK4/6i—PFS 6.5 months	Phase I CBR 55%, ORR 8.0%	[48]
**Camizestrant (AstraZeneca)**	Phase II SERENA-2	50%	Phase III SERENA-4 (NCT04711252) CDK4/6i = palbociclib Phase III SERENA-6 (NCT04964934) CDK4/6i = palbociclib	NA	Phase II SERENA-2—7.2 months (75mg) in patients with prior CDK4/6i compared to 3.7 months on fulvestrant	Phase I CBR of 42.3%, ORR 16.3% Phase II SERENA-2 (75 mg) CBR 47%, ORR 16%	[49,51,52]
**Ritodestrant (G1 Therapeutics)**	Phase I	69%	Phase II (NCT03455270) CDK4/6i = palbociclib	NA	NA	Phase I CBR 28%	[47]

PFS = progression-free survival, ORR = overall response rate, CBR = clinical benefit rate, NA = not applicable. NCT included for actively recruiting trials.

**Table 4 cancers-15-01855-t004:** Novel agents post CDK4/6i. MOA = mechanism of action, PFS = progression-free survival, CBR = clinical benefit rate, WT= wild type.

Drug	ARV-471	Capivasertib	H3B-6545	OP-1250	Samuraciclib plus Fulvestrant	Enobosarm	Erdafitinib
**MOA**	Proteolysis targeting chimeras (PROTAC)	AKT pathway inhibitor	Selective estrogen receptor covalent antagonist (SERCA)	Complete estrogen receptor antagonists (CERAN)	Selective inhibitor of CDK7	Selective androgen receptor modulator	Fibroblast growth factor receptor (FGFR) Inhibitors
**Phase**	I/II	III	I/II	I	I/II	II	II
** *n* **	71	355	94	27	31	50	48
**Prior CDK4/6i**	100%	69%	85%	100%	100%	NA	NA
**Adverse Events (>10%)**	Fatigue (21%) Arthralgia (13%) Nausea (11%) Hot flush (10%)	Diarrhea (72.4%) Nausea (34.6%) Rash (22.0%) Fatigue (20.8%) Vomiting (20.6%) Headache (16.9%) Decreased appetite (16.6%) Hyperglycemia (16.3%) Stomatitis (14.6%) Asthenia (13.2%) Pruitus (12.4%) Anemia (10.4%) UTI (10.1%)	Anemia (19%) Nausea (17%) Fatigue (16%) Diarrhea (12%) Creatinine clearance decrease (39%) Hemoglobin decrease (38%) Lymphocytes decrease (37%) ALT increase (14%) AST increase (13%) Bilirubin increase (12%) Creatinine increase (12%) Bradycardia (35%)	Nausea (59.5%) Fatigue (35.1%) Constipation (24.3%) Headache (24.3%) Vomiting (24.3%) Decreased appetite (21.6%) Neutropenia (18.9%) Rash (16.2%)	Diarrhea (90%) Nausea (81%) Vomiting (74%) Fatigue (36%) Decreased appetite (29%) Abdominal pain (23%) AST increased (13%) Dysgeusia (13%) Headache (13%) Upper abdominal pain (13%)	Improvement in measurements including mobility Anxiety/depression Pain	Dry mouth (43%) Fatigue (39%) Anorexia (27%) Alopecia (24%) Oral mucositis (24%) Nausea (24%) Constipation (22%) Dry eye (22%) Vomiting (22%) Diarrhea (20%) Dysgeusia (20%) Eye disorders (20%) Anemia (18%) Weight loss (14%) Blurred vision (12%) Increased ALP (18%) Leukopenia (12%) Increased Cr (12%)
**PFS**	3.7 months	7.2 months	5.1 months	NA	TP53 WT: 32 weeks TPS mutant: 7.9 weeks	5.6 months	3.4 months
**Response rate**	CBR 38%		CBR 40.3% ORR 16.7%	CBR 21% ORR 9%	CBR 36%	AR ≥ 40% CBR = 80% AR < 40% CBR = 18%	6-month PFS rate was 15%
**References**	[42]	[57]	[58,59]	[60]	[61]	[62,63]	[64]

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
