# Peer review of "Post-CDK 4/6 Inhibitor Therapy: Current Agents and Novel Targets"

_cancers, 2023, doi:10.3390/cancers15061855_

Round 1

Reviewer 1 Report

Well-written and comprehensive. No major edits!

I would suggest making sure to describe what each drug does mechanistically. For instance, Alpelisib is a PI3K inhibitor. Should be included somewhere near line 98. Fulvestrant is a SERD.....Everolimus....exemestane....etc

Line 55 is missing the word 'benefit' before 'survival', I believe. 

Author Response

Thank you for your time and consideration! I will add MOA to each drug section, and fix the missing word you highlighted. 

Reviewer 2 Report

This review summarized the emerging targets and therapeutics that may be approved in the future for the treatment of HR+, HER- mBC, which could be accepted for publication in Cancers after addressing the following problmes

1. The antitumor mechanism of key drugs mentioned in this review should be listed, better in a table.

2. The structures of the  key drugs mentioned in this review should also be added.

Author Response

1. In table 4, MOA is included in the title bar of the table for all drugs without FDA approval of at least one drug in its class.  I have changed the order so that MOA is below the name of the drug. 

2. Since this paper is more clinically oriented, I did not include drug structures.  We wouldn't likely have the conversational space to discuss or review drug structures in a paper such as this one. 

Reviewer 3 Report

Excellently written manuscript.

Minor comments:

1. shorter description of the use of chemotherapy after progression to CDK 4/6 would be beneficial but  not necessary

2. It is necessary to correct the structure of the tables where there are unordered indents.

Author Response

Thank you for your comments! I have taken your comments and your fellow reviewers comments in consideration when reviewing the section on approved second line therapies.  I have addressed formating problems with the table. 

Round 2

Reviewer 2 Report

No